# The bZIP transcription factor AREB3 mediates FT signalling and floral transition at the Arabidopsis shoot apical meristem

**Damiano Martignago**[1�], **Vítor da Silveira Falavigna**[2�], **Alessandra Lombardi**[1], **He Gao**[2], **Paolo Korwin Kurkowski**[1], **Massimo Galbiati**[1¤], **Chiara Tonelli**[1], **George Coupland**[2], **Lucio Conti**[1]*

**1** Dipartimento di Bioscienze, Università degli Studi di Milano, Milan, Italy, **2** Max Planck Institute for Plant Breeding Research, Cologne, Germany

උ These authors contributed equally to this work.
¤ Current address: Istituto di Biologia e Biotecnologia Agraria, Consiglio Nazionale Delle Ricerche, Milan, Italy
* lucio.conti@unimi.it

**Data Availability Statement:** All relevant data are within the manuscript and its Supporting Information files.

## Abstract

The floral transition occurs at the shoot apical meristem (SAM) in response to favourable external and internal signals. Among these signals, variations in daylength (photoperiod) act as robust seasonal cues to activate flowering. In Arabidopsis, long-day photoperiods stimulate production in the leaf vasculature of a systemic florigenic signal that is translocated to the SAM. According to the current model, FLOWERING LOCUS T (FT), the main Arabidopsis florigen, causes transcriptional reprogramming at the SAM, so that lateral primordia eventually acquire floral identity. FT functions as a transcriptional coregulator with the bZIP transcription factor FD, which binds DNA at specific promoters. FD can also interact with TERMINAL FLOWER 1 (TFL1), a protein related to FT that acts as a floral repressor. Thus, the balance between FT-TFL1 at the SAM influences the expression levels of floral genes targeted by FD. Here, we show that the FD-related bZIP transcription factor AREB3, which was previously studied in the context of phytohormone abscisic acid signalling, is expressed at the SAM in a spatio-temporal pattern that strongly overlaps with FD and contributes to FT signalling. Mutant analyses demonstrate that AREB3 relays FT signals redundantly with FD, and the presence of a conserved carboxy-terminal SAP motif is required for downstream signalling. AREB3 shows unique and common patterns of expression with FD, and *AREB3* expression levels are negatively regulated by FD thus forming a compensatory feedback loop. Mutations in another bZIP, *FDP*, further aggravate the late flowering phenotypes of *fd areb3* mutants. Therefore, multiple florigen-interacting bZIP transcription factors have redundant functions in flowering at the SAM.

## Author summary

In this research, we studied how plants regulate the time to flower. This process is highly sensitive to the environment, including seasonal changes in day length. In *Arabidopsis*

**Funding:** This work was supported by a Research Grant from the HFSP Ref.-No: RGP0011/2019 and from the University of Milan - SEED 2019 - DISENGAGE Ref. no.1236 to LC. DM is supported by a research fellowship co-funded by the European Union - ESF, REACT-EU, PON Ricerca e Innovazione 2014-2020. This work was also funded by the Deutsche Forschungsgemeinschaft through Cluster of Excellence CEPLAS (EXC 2048/1 Project ID: 390686111) to GC, and the laboratory of GC receives core funding from the Max Planck Society. VSF was supported by the European Union's Horizon 2020 research and innovation programme under the Marie Skłodowska-Curie grant agreement 894969 and the von Humboldt Foundation (BRA 1210514 HFST-P). The authors wish to acknowledge the support of the APC central fund of the University of Milan. The funders had no role in study design, data collection and analysis, decision to publish, or preparation of the manuscript.

**Competing interests:** The authors have declared that no competing interests exist.

*thaliana* long day conditions, typical of spring and summer, stimulate the production of a specialised protein signal called florigen in leaves to initiate flowering at the shoot (the growing tip of the plant). Florigen proteins move long distance to the shoot where they interact with another set of proteins, the best known of which is FD, belonging to group A bZIPs. Here, we discovered that all group A bZIP proteins can also bind to the florigen. We also found that the bZIP AREB3, present at the shoot like FD, when mutated together with *FD* caused an aggravated delay in flowering compared with single mutant plants. Mutations in a third bZIP, *FDP*, resulted in an even later flowering compared with double mutants, showing that the effect of these mutations is cumulative. It appears that the more of this family of regulatory proteins we remove, the later the plant flowers. In conclusion, we discovered that many more bZIPs than previously thought can interact with florigens to regulate flowering time.

## Introduction

Many plant species detect variations in daylength (photoperiod) and in response to these align their growth and development to the most beneficial environmental conditions. *Arabidopsis thaliana* responds to long days (LDs), typical of spring/summer at temperate latitudes, to activate flowering and initiate its reproductive cycle [1,2]. Extensive mutagenesis screens led to the definition of a genetic pathway and transcriptional cascade activated by LDs, and major components of this pathway are conserved across species. Photoperiodic flowering involves the transmission of signals from the leaves–the site of photoperiod perception–to the shoot apical meristem (SAM)–where the floral transition and floral development occur. FLOWERING LOCUS T (FT) acts as the main systemic florigenic signal, being produced in the leaf vasculature in response to LDs and moving to the SAM [3–6]. In the SAM, FT triggers extensive transcriptional reprogramming, ultimately causing a change in the identity of lateral organ primordia that switch from forming leaves and axillary branches to forming flowers [7–9].

FT belongs to the phosphatidylethanolamine-binding proteins (PEBPs) superfamily, which includes structurally, but not functionally, related proteins described from bacteria to humans [10–12]. In plants, PEBPs are usually regarded as transcriptional coregulators [13]. Crystallographic data derived from rice florigen Hd3a describes nuclear-localised hexameric florigen activation complexes (FAC), consisting of pairs of Hd3a proteins, scaffold 14-3-3 proteins, and bZIP (basic leucine zipper) transcription factors (TFs) [14]. Several independent studies support a general model in which phosphorylated bZIP TFs provide DNA binding selectivity, whereas florigens stimulate transcription at target promoters, possibly stabilizing the formation of the bZIP–DNA complex [15–18]. Phosphorylation of a conserved 14-3-3 binding site at the C terminus of the bZIP TF FD, called SAP motif (RXX(pS/pT)XP), has been described as essential for the formation and function of the FAC [14]. Although non-phosphorylatable versions of the FD SAP motif showed the impaired formation of the FAC complex, FD still binds to DNA *in vivo*, even in the absence of FT [14,15,17]. FD activates the expression of and directly binds to many flowering-time genes, including *SUPPRESSOR OF OVEREXPRESSION OF CONSTANS 1* (*SOC1*) and *FRUITFULL* (*FUL*), and floral-meristem identity genes such as *APETALA1* (*AP1*) and *LEAFY* (*LFY*) [8,19–23], and FT is proposed to enhance FD binding to its target genes [15]. Another PEBP, TERMINAL FLOWER 1 (TFL1), is present at the SAM and antagonises FT function, perhaps by competing for the binding to FD [21,24,25]. The TFL1–FD complex formation at FD target chromatin mostly results in transcriptional repression [18,21]. Thus, FD function is key for the assembly of different PEBP complexes at target

DNA sequences, causing different transcriptional fates at regulated genes, and ultimately affecting the flowering process.

While FD plays a key role in mediating FT signalling at the SAM, *fd* mutants only partially suppress the early flowering conferred by overexpression of *FT* [19,26]. This suggests that other genes are also involved in the FT-mediated regulation of flowering. Indeed, some degree of functional redundancy between *FD* and *FD PARALOGUE* (*FDP*) exists, although *fd fdp* double mutants still retain substantial flowering responsiveness to LDs [26,27]. Other potentially redundant functions to *FD* may lie within the evolutionarily-related group A bZIP TFs. This group includes several proteins described as mediators of abscisic acid (ABA) signal transduction [28,29], such as ABA INSENSITIVE 5 (ABI5) and related ABRE-binding (AREB) proteins or ABRE-binding factors (ABFs), which were characterised by their common binding to conserved ABA-responsive elements (ABREs, PyACGTGG/TC) [30,31]. Recent studies indicate that ABI5 and TFL1 proteins act in the same protein complex to control seed size and germination [32], suggesting that other group A bZIP TFs can associate with PEBPs to control different traits. The group A bZIPs ABF3 and ABF4 promote flowering from the leaves in response to drought [33]. However, it is currently unknown if, besides FD and FDP, other group A bZIPs play any role in relaying FT signalling at the SAM. Here, we studied potential interactions between group A bZIP TFs and the PEBP FT and TFL1. By using CRISPR-Cas9-based mutagenesis and genetics approaches, we demonstrate that AREB3 is a novel interactor of FT, acting redundantly with FD and FDP in flowering-time regulation at the SAM. Confocal microscopy imaging of shoot meristems reveals a striking overlap between AREB3 and FD expression, supporting their redundant role. Notably, *AREB3* expression levels are negatively regulated by FD, so higher levels of *AREB3* mRNA and its encoded protein are observed in *fd* mutants. Our results contribute to increasing knowledge of compensatory mechanisms between proteins that play a key role in the photoperiodic regulation of flowering and show that more bZIP TFs than previously known are expressed at the SAM and can interact with FT to finely regulate floral transition.

## Results

### Widespread interactions between Arabidopsis group A bZIPs and FT or TFL1

To gain insights into the potential interactions between group A bZIPs and FT or TFL1 proteins, a yeast two-hybrid (Y2H) assay was established comprising all group A bZIP sequences fused to the activation domain. This assay confirmed robust interactions between FT/TFL1 and FD/FDP [19,20]. Notably, FT and TFL1 also interacted with nearly all other group A bZIP TFs (Figs 1A and S1). These results suggest that FT can interact with a wider range of group A bZIP TFs than previously proposed and that some of these interactions might contribute to inducing floral transition at the SAM redundantly with FD and FDP.

Next, we screened the group A bZIP TFs for the presence of a putative SAP motif in their C terminus. Mode I canonical RXX(pS/pT)XP motifs were described as highly conserved 14-3-3 binding sites, with plants commonly presenting an extended LX(R/K)SX(pS/pT)XP motif [34]. We found that most group A bZIP genes presented at least one splicing form encoding a canonical SAP motif, with a few exceptions at the conserved proline residue (Fig 1B, [28]). In Y2H and EMSA assays, this motif is critical for FD interaction with FT (Fig 1C, [15,19]). Among all group A bZIP TFs, AREB3 (also known as DPBF3, AtbZIP66, At3g56850) has the most similar SAP motif to FD and FDP. Its proposed SAP motif contains a potentially phosphorylatable serine (S294) instead of the threonine of FD (T282) and FDP (T231). By screening publicly available proteomic datasets (S1 Table, [35]), we found that AREB3 is phosphorylated at S294 as well as at other ABA-related sites [36–41].

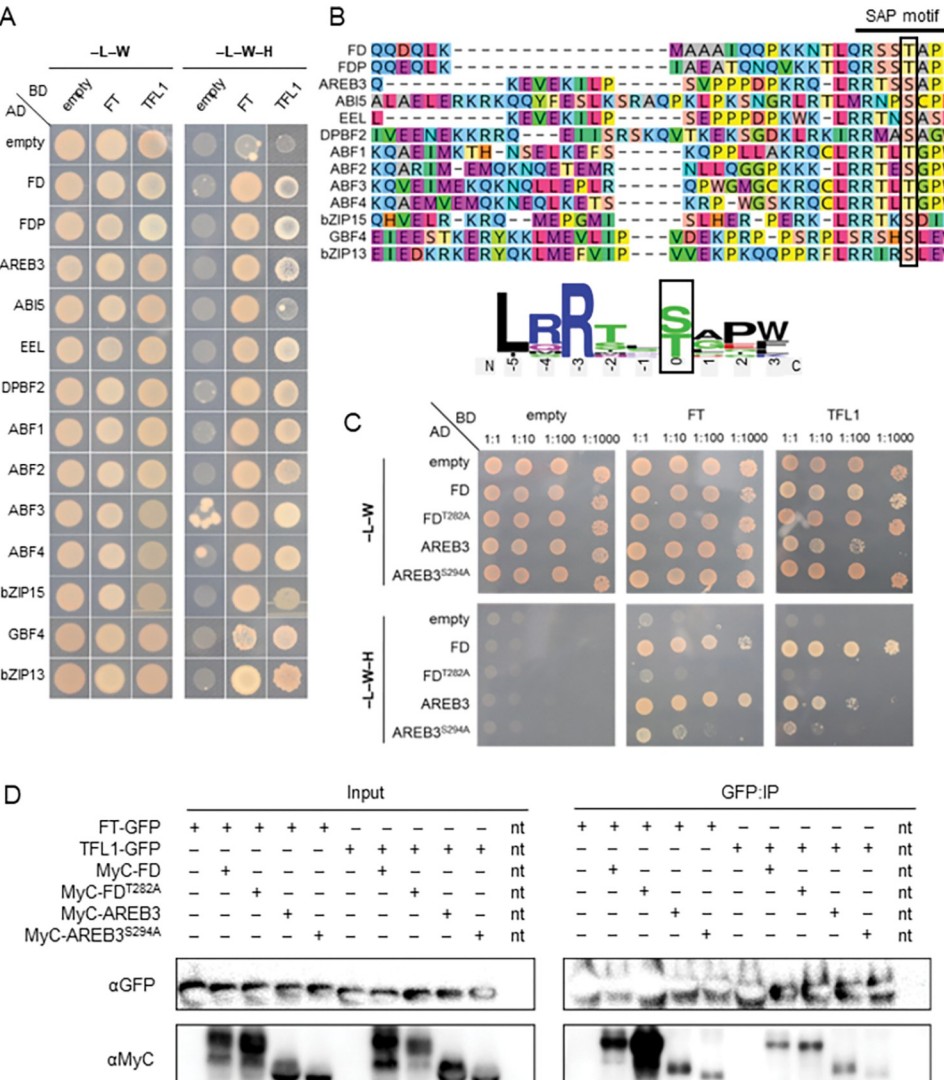

**Fig 1. FT and TFL1 interact with several group A bZIP TFs.** (A) Y2H assays were conducted to test protein interactions among group A bZIP TFs and the PEBP proteins FT and TFL1. See S1 Fig for further information. (B) Protein alignments of the C-terminal region of the group A bZIP TFs show strong conservation of the RXX(pS/pT)XP SAP motif. The phosphorylatable T/S residue, which is T282 and S294 for FD and AREB3, respectively, is boxed. The consensus sequence of the SAP motif is depicted in the logo format (made with weblogo.berkeley.edu). (C) Y2H assays of protein interactions among wt FD, wt AREB3, FD$^{T282A}$ and AREB3$^{S294A}$ with FT and TFL1. (D) *N. benthamiana* co-IP of protein interactions among wt and mutated versions of FD and AREB3 with FT and TFL1. Protein–protein interactions were tested in pairs by co-agroinfiltration of tobacco leaves. FD and AREB protein versions were translationally fused to MyC, whereas both PEBPs were translationally fused to GFP. The input was composed of total proteins recovered before the IP. GFP-fused proteins were pulled down using anti-GFP nanobody (VHH) beads and immunoblotted using α-MyC or α-GFP antibodies. Additional controls are present in S3 Fig. Nt, non-transformed.

## Mutations in the SAP motif do not abolish the interaction of bZIP TFs with FT/TFL1 in plant cells

We tested the importance of the phosphorylatable S294 residue for FT interaction by constructing the AREB3$^{S294A}$ mutant and found that its interaction with either FT or TFL1 is weakened in a Y2H assay (Fig 1C). Similar results were obtained with AREB3$^{ΔSAP}$, a truncated version of AREB3 (R291*) lacking the SAP motif cds (S2 Fig). The interaction between AREB3

and FT/TFL1 was further verified through co-immunoprecipitation (co-IP) of transiently expressed, epitope-tagged versions of AREB3 and FT/TFL1 in *Nicotiana benthamiana* (Fig 1D). As expected, both FD and AREB3 proteins were co-purified with FT and TFL1, supporting theirs *in planta* interaction. In this assay, the AREB3$^{S294A}$ and FD$^{T282A}$ mutant proteins were also co-immunoprecipitated with FT and TFL1. Similar results were obtained in another tobacco co-IP assay, in which FD and FD$^{\Delta SAP}$ but not CONSTANS (CO) interacted with FT (S3 Fig). Next, bimolecular fluorescence complementation (BiFC) assays confirmed comparable levels of fluorescence reconstitution in nuclei (n>150) upon co-expression of wild-type (nYFP:AREB3) or truncated (nYFP:AREB3$^{\Delta SAP}$) versions of AREB3 with FT:cYFP in *N. benthamiana* (S4 Fig). These results show that, in these transient assays, FT and TFL1 interaction with AREB3 and FD can occur in plant cells independently of the SAP motif.

## Functional redundancy between FD and AREB3

Loss of function *areb3-1* T-DNA insertional mutants did not reveal significant alterations in flowering time compared with wild-type plants grown under LD or short-day (SD) conditions (Fig 2A and 2B). To test for functional redundancy between *AREB3* and *FD*, we generated *areb3-1 fd-3* and *areb3-1 fd-4* double mutants. Under LDs, these plants flowered significantly later than *fd-3* and *fd-4* single mutants as determined by the total number of rosette leaves formed (Fig 2A) and time of bolting (S5 Fig). No delay in flowering time was observed in *areb3-1 fd-3* mutants under SDs (Figs 2B and S5), indicating that the role of *AREB3* in floral promotion was LD-specific.

Insertional *areb3-1* and *areb3-2* mutants were found to contain the same T-DNA insertion in the last intron and retain residual expression of full-length *AREB3* transcripts (S6 and S7 Figs). Similarly, *AREB3* transcripts were also detected in *areb3-1 fd-3* and *areb3-1 fd-4* mutants (S7 Fig). New alleles of *AREB3* (named *areb3-Cr*) were generated in the *fd-3* genetic background by CRISPR-Cas9-mediated mutagenesis of a DNA region immediately upstream of the SAP motif-encoding sequence (S8 Fig). This strategy allowed direct assessment of the activity of AREB3 proteins lacking a SAP motif in the absence of a functional FD. Several independent allelic combinations of *AREB3* were isolated in the T1 generation (S8 and S9 Figs). The *areb3-Cr1* and *areb3-Cr2* alleles caused 1-bp frameshift insertions upstream of the SAP motif and therefore were predicted to delete the motif (+T for *areb3-Cr1*, +A for *areb3-Cr2*, Fig 2C). Similarly, disruption of the encoded AREB3 SAP motif sequence was also obtained by a single nucleotide deletion (–C for *areb3-Cr4*). Under LDs, *fd-3 areb3-Cr1*, *fd-3 areb3-Cr2* and *fd-3 areb3-Cr4* flowered significantly later than *fd-3* but did not present flowering-time alterations under SDs (Figs 2B, 2D, 2E and S5). Conversely, the *areb3-Cr3* allele, presenting a homozygous in-frame 6-bp deletion immediately upstream of the AREB3 SAP motif, showed no additive effect on the flowering time of *fd-3* (Fig 2D and 2E). In the wild-type background, *areb3-Cr1* and *2* single mutants displayed a mild late flowering phenotype (S9 Fig). These results support the idea that the integrity of the SAP motif is required for the promotion of flowering, and that amino acids between the SAP motif and the bZIP DNA-binding domain can be removed without affecting flowering.

In whole seedlings, the accumulation of *AREB3* full-length transcripts was not altered in *areb3-Cr1/3 fd-3* mutants compared with the wild type (S7 Fig). Therefore, the observed late-flowering phenotypes conferred by these alleles were not due to nonsense-mediated *AREB3* mRNA decay. We performed a similar CRISPR approach on the encoded SAP motif of FD to test its functional importance *in vivo*. A mutant line carrying a single nucleotide insertion (+A, *fd-Cr1*) disrupting the SAP motif was isolated (Fig 2F). The *fd-Cr1* line flowered significantly later than wild type and similarly to the *fd-3* strong T-DNA insertional mutant (Fig 2G).

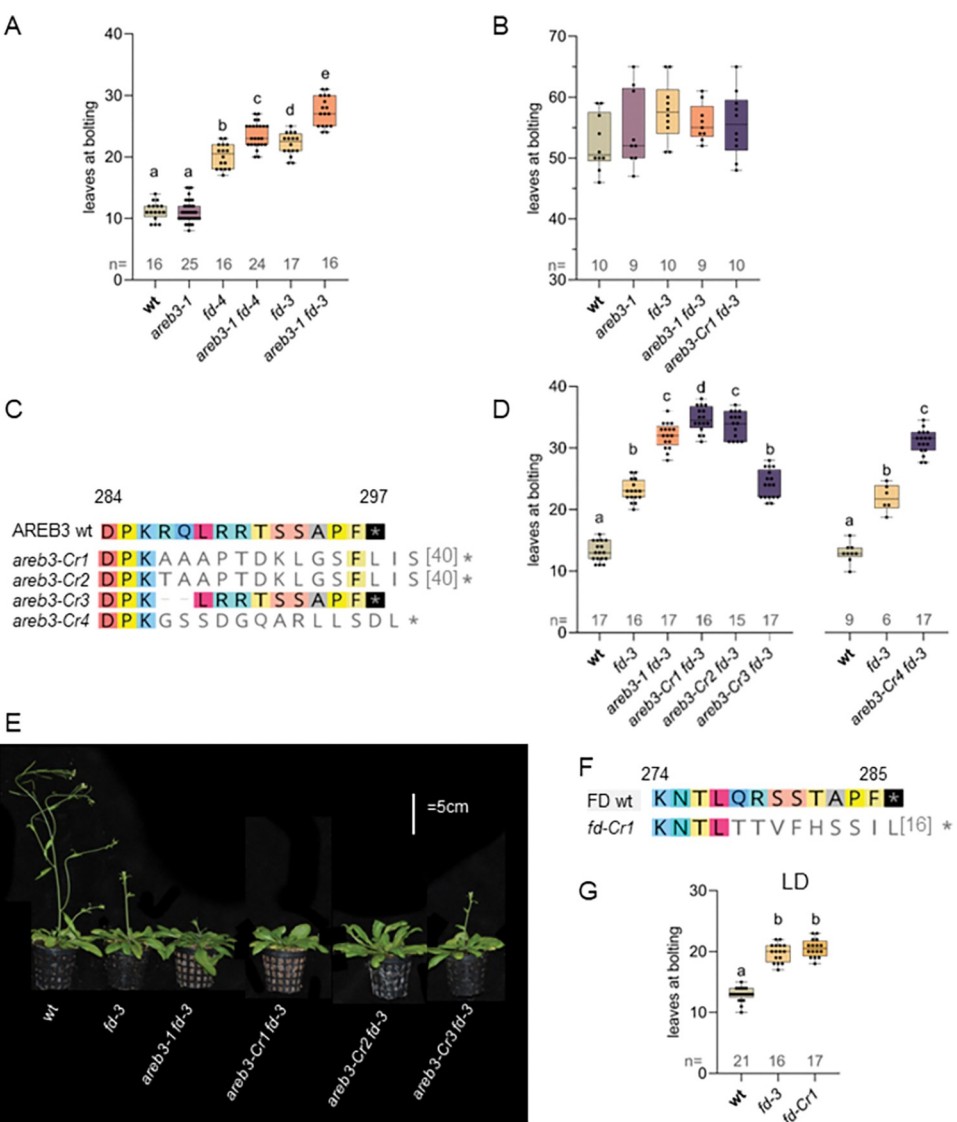

**Fig 2. Genetic redundancy between *AREB3* and *FD*.** (A) Flowering time of the indicated genotypes under LDs (significance a *vs* b, c, d, e p<1e-15; b *vs* c p = 3.42e-6; d *vs* e p = 2.83e-11; b *vs* d ns;). (B) Flowering-time analysis of the indicated genotypes under SDs. (C) Alignment of the predicted C-terminal protein sequences of AREB3 mutants obtained using CRISPR-Cas9 compared to the wt (starting from D284). (D) Flowering time of *areb3* CRISPR mutants in the *fd-3* background under LDs (significance a *vs* b, c, d p<4.84e-10; b *vs* c, d p = 4.84e-10; c *vs* d p = 0.004). Analysis of the *areb3-Cr4* allele is shown on the right (significance a *vs* b p = 2.51e-9; b *vs* c p = 1.58e-10). (E) The phenotype of the indicated genotypes at 6 weeks after sowing under LDs. (F) Alignment of the predicted C-terminal protein sequences of FD mutant obtained using CRISPR-Cas9 compared to the wt (starting from K274).(G) Flowering time of *fd-Cr1* mutants under LDs (significance a *vs* b p<1e-15).

## AREB3 distribution at the SAM overlaps with FD

The stronger late-flowering phenotype observed in *fd-3 areb3* mutants compared to *fd-3* may be due to *AREB3* partially compensating for the loss of *FD* at the SAM. To test whether AREB3 is expressed in a similar temporal and spatial pattern to FD, stable transgenic lines expressing a fusion of AREB3 to the VENUS fluorescent protein (*pAREB3:VENUS:AREB3*) were constructed in the *areb3-1* background. Four independent homozygous T3 lines were obtained, and RT-PCR experiments confirmed the expression of the mRNA of the chimeric version of

*AREB3* at similar levels to the endogenous gene (S10 Fig). Confocal laser microscopy imaging of shoot apices of *pAREB3:VENUS:AREB3 areb3-1* (#11.4) plants revealed that AREB3 was detectable in most cells of the vegetative SAM, including the L1, L2 and L3 meristematic cell layers, and young leaf primordia (Fig 3A). After the floral transition, AREB3 was also present

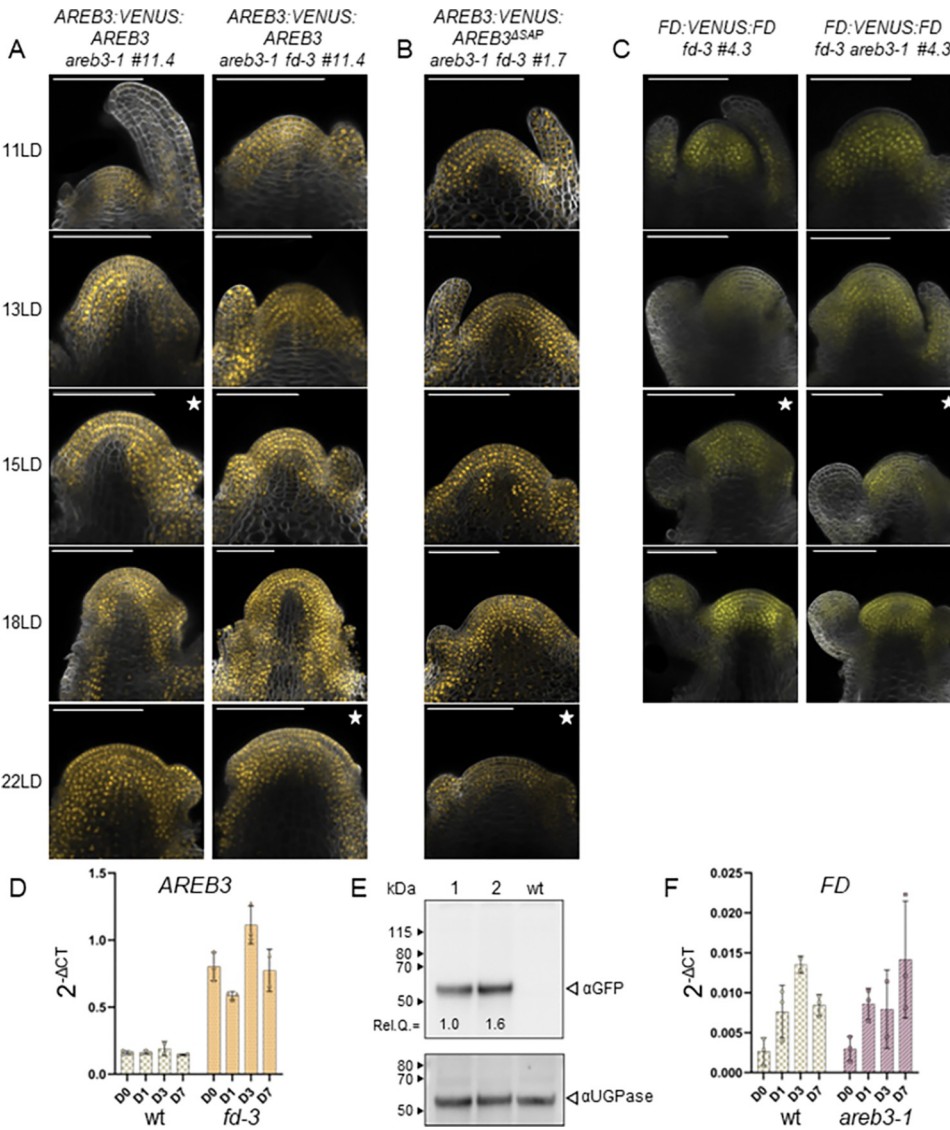

**Fig 3. Gene expression and protein localisation of AREB3 and FD.** (A) Confocal analysis of dissected apices during the floral transition of *pAREB3:VENUS:AREB3* in *areb3-1* and *areb3-1 fd-3* backgrounds. (B) Dissected apices of *pAREB3:VENUS:AREB3^{ΔSAP}* in *areb3-1 fd-3* were analysed by confocal microscopy. (C) Confocal analysis of dissected apices of *pFD:VENUS:FD* in *fd-3* and *fd-3 areb3-1* backgrounds. In these assays, at least three apices of each line were analysed and a representative image of the spatial distribution of the proteins was selected. The star indicates the establishment of the inflorescence meristem. Scale bars, 100 μm. (D) mRNA levels of *AREB3* in wt and the *fd-3* mutant in SAM-enriched tissue. (E) AREB3:VENUS protein abundance in SAMs of *areb3-1* (lane 1) and *areb3-1 fd-3* (lane 2) isogenic lines #11.4. Upper panel: immunoblot analysis of AREB3:VENUS detected with αGFP antibodies; Rel. Q. indicates protein quantity relative to the sample loaded in lane1. Lower panel: immunoblot analysis of UGPase, used as a loading control. For each panel, numbers indicate molecular weights. (F) mRNA levels of *FD* in wt and *areb3-1* mutant in SAM-enriched tissue. In both experiments, Arabidopsis plants were grown for two weeks in short-day (SD) conditions and then shifted to long days (LD). Samples were harvested at ZT8 before the shift (D0) and 1, 3, and 7 days (D1, D3, and D7, respectively) after the shift to flowering-inducing photoperiod. Each point represents an independent pool of around five meristems. The experiment was performed twice with similar results.

throughout the inflorescence meristem, in young flower primordia and stems. One representative *pAREB3*:*VENUS*:*AREB3* line (#11.4) was crossed with *areb3-1 fd-3* to obtain *pAREB3*: *VENUS*:*AREB3 areb3-1 fd-3*. The confocal analysis did not reveal obvious changes in the AREB3 spatial distribution at the SAM in the *areb3-1 fd-3* background in comparison to *areb3-1*, despite their different developmental stages (Fig 3A). Still, the insertion of *AREB3*: *VENUS*:*AREB3* into *areb3-1 fd-3* double mutants complemented the later-flowering phenotype caused by *areb3-1*, indicating that the VENUS:AREB3 fusion protein was functional (S10 Fig). In contrast, Basta-resistant *areb3-1 fd-3* T2 transgenic lines carrying a mutant construct lacking the SAP motif of AREB3 (*pAREB3*:*VENUS*:*AREB3*$^{\Delta SAP}$) flowered as late as *areb3-1 fd-3* (S10 Fig), supporting the importance of the AREB3 SAP motif for floral promotion. To understand if the lack of the SAP motif could influence AREB3 protein accumulation or subcellular localisation at the SAM, we analysed homozygous T3 *pAREB3*:*VENUS*:*AREB3*$^{\Delta SAP}$ (#1.7) plants at different developmental stages. The VENUS:AREB3$^{\Delta SAP}$ protein was nuclear localised and expressed in a similar spatio-temporal pattern to that of the wild-type VENUS: AREB3 protein during floral transition (Fig 3B). Thus, while an FT–AREB3$^{\Delta SAP}$ complex could potentially form at the SAM, no FT signalling is elicited in the absence of the SAP motif.

We next asked whether changes in FD function might influence *AREB3* accumulation or *vice versa*. Wild-type, *areb3-1* or *fd-3* plants were shifted from SDs to LDs to activate flowering, and the transcript levels of *AREB3* or *FD* were assayed by RT-qPCR from manually dissected shoot apices. Notably, *AREB3* mRNA levels were higher in *fd-3* mutants than in wild-type plants (Fig 3D). In agreement, immunoblot analysis of shoot apices from *pAREB3*:*VENUS*: *AREB3* in *areb3-1* or *areb3-1 fd-3* backgrounds collected 3 days after the shift to LDs revealed an increase (1.6X) in VENUS:AREB3 protein accumulation in *fd-3* compared to the isogenic *FD* background (Fig 3E). Thus, the increase in *AREB3* transcript levels detected in *fd* mutants translates into more protein accumulation in this genetic background. Comparable transcript levels of *FD* were identified between *areb3-1* mutants and wild-type plants at all time points analysed (Fig 3F). Similar to AREB3 (Fig 3A), no clear changes in the spatial distribution of FD at the SAM were observed when comparing *fd-3* to *fd-3 areb3-1* (Fig 3C). These results suggest that the *AREB3* upregulation in *fd* mutants may partially compensate for the effect of loss of FD activity on flowering time.

For this compensation to occur, FD and AREB3 proteins should share similar spatial and temporal localisation during floral transition. To test this possibility, transgenic lines expressing a fusion of FD to the mCHERRY fluorescent protein (*pFD*:*mCHERRY*:*FD*) were constructed in the *fd-3* background, and three homozygous single-copy lines were obtained. These lines complemented the late-flowering phenotype of *fd-3* and showed similar spatial and temporal localisation to VENUS:FD [26]. Next, double hemizygous *pFD*:*mCHERRY*:*FD pAREB3*: *VENUS*:*AREB3 fd-3 areb3-1* lines were then examined and showed a strong overlap in the accumulation of mCHERRY:FD and VENUS:AREB3 at the SAM (Fig 4). Yet, AREB3 was consistently identified in the L1 meristematic layer, young flower primordia and developing stems, regions in which FD is absent [7,26]. These results suggest that the partially redundant genetic relationship between *AREB3* and *FD* may be due to their encoded proteins having overlapping spatial patterns of expression and related biochemical functions in the SAM.

## Redundancy across three group A bZIPs in mediating FT signalling

Group A bZIP TFs are implicated in different aspects of flowering-time regulation, upstream and downstream of *FT* [33]. None of the mutant combinations between *areb3* and *fd* displayed obvious changes in *FT* transcript accumulation in leaf tissues compared with the wild type (S11 Fig). Therefore, the flowering-time defects of *areb3 fd* mutants appear to derive from

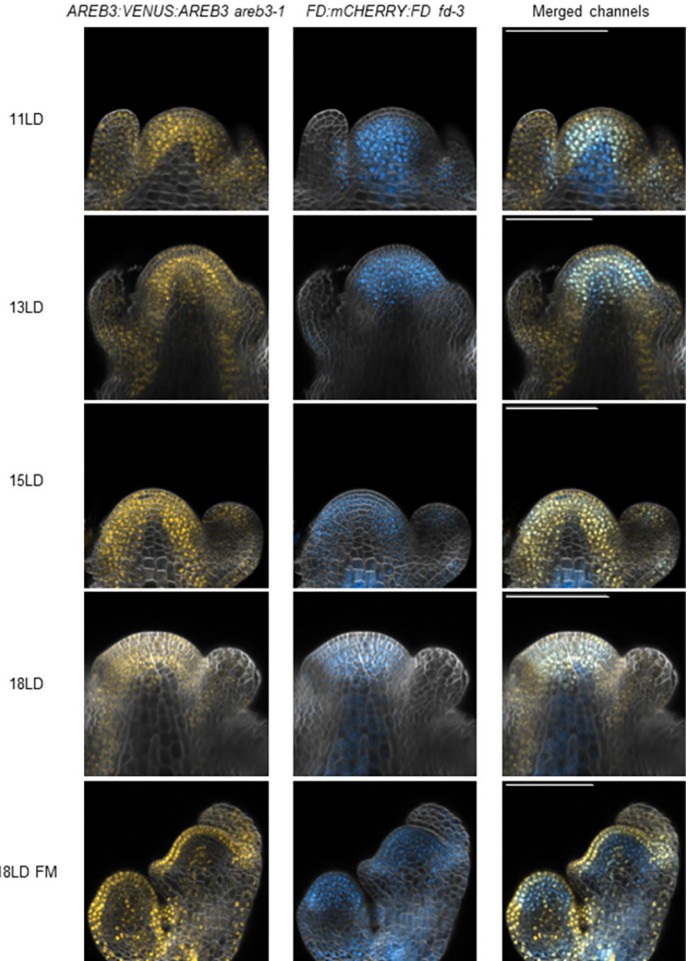

**Fig 4. Colocalisation analysis of FD and AREB3 in the SAM.** *pAREB3*:*VENUS*:*AREB3 areb3-1* and *pFD*:*mCHERRY*: *FD fd-3* plants were crossed and the F1 progeny was analysed by confocal microscopy. Dissected apices were sampled at 11, 13, 15 and 18 long days (LD) and simultaneously imaged for VENUS, mCHERRY and Renaissance 2200 fluorescence. FM, floral meristem. Scale bars, 100 μm.

their effects in the SAM, downstream of *FT* transcriptional activation. To test this hypothesis further, *areb3* and *fd* mutations were introduced into a transgenic background overexpressing *FT* under the control of the *Gas1* (*Galactinol Synthase 1*) promoter, which is active specifically in the phloem companion cells of the minor veins of leaves and confers an extreme early-flowering phenotype (Fig 5A and 5B) [3,42]. As expected, the early-flowering phenotype of *pGas1*: *FT* was reduced in the *fd-3* background, and even more so in double mutants of *areb3-1 fd-3* (Fig 5A and 5B). *pGas1:FT areb3-1 fd-3* plants were later flowering than wild type, but still earlier flowering than *areb3-1 fd-3* mutants (Fig 5A and 5B). Therefore, besides FD and AREB3, other TFs likely mediate FT signalling at the shoot apex, and FDP had already been proposed to perform such a role [4,26]. We found that *fdp* mutations did not alter the flowering time of *areb3* mutants, but both *fdp* (*-CRP3* or *-CRP2* alleles) or *areb3*-1 significantly enhanced the late-flowering phenotype of *fd* mutants (Figs 5C and S12, [26]). Moreover, an additive delay in flowering time was observed in triple *areb3-1 fd-3 fdp-CRP* mutants compared with double mutant combinations (Figs 5C and S12). These data indicate genetic redundancy among *FD*, *AREB3* and *FDP* in promoting floral transition.

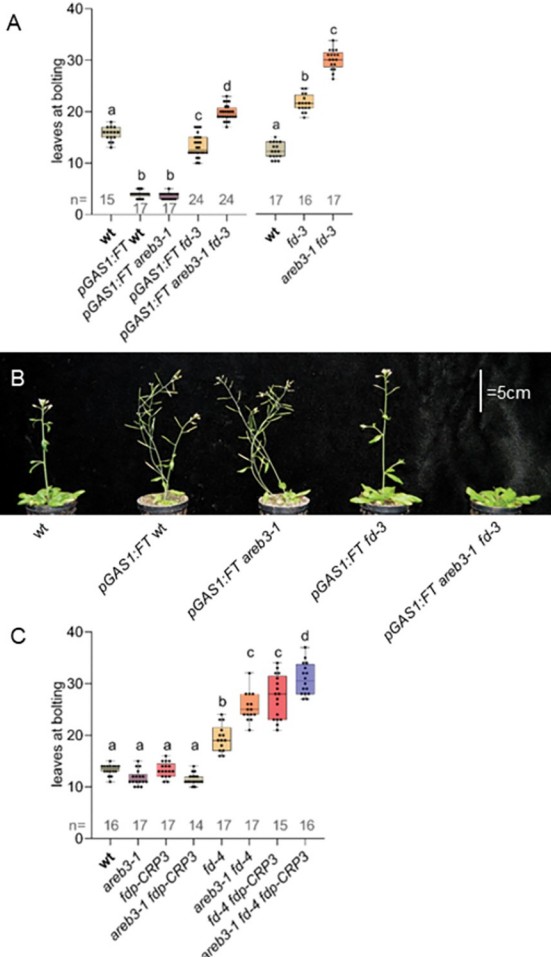

**Fig 5. Genetic interaction between *AREB3*, *FT*, *FD*, and *FDP*.** (A) Flowering time of the indicated genotypes under LDs (significance a *vs* b p = 4.82e-10; b *vs* c, d p = 4.82e-10; c *vs* d p = 4.82e-10). Right panel: flowering time of the indicated genotypes which were used as controls (shared with Fig 2B); significance a *vs* b, c p<4.84e-10; b *vs* c p = 4.84e-10). (B) The phenotype of the indicated genotypes under LDs at 4 weeks after sowing. (C) Flowering time of the indicated genotypes under LDs (significance a *vs* b, c, d p<8.93e-09; b *vs* c p = 3.33e-8; c *vs* d p<2.03e-03).

## TFL1 signalling antagonises FT partially through AREB3

TFL1 antagonises FT signalling by competing for interaction with FD [21]. Our genetic and biochemical data indicate that AREB3 also interacts with TFL1 and FT similarly to FD (Figs 1 and 2). The absence of *TFL1* leads to increased activity of FT and FD, resulting in early flowering [27]. The *tfl1* mutants are also characterised by a determinate inflorescence architecture, often lacking cauline leaves and subtending paraclades (I1 phase), and show a reduced number of floral buds on the main shoot (I2) [43,44]. In line with previous observations, the flowering-time defects of *tfl1* plants were largely suppressed in *fd-3 tfl1-18* double mutants (Fig 6A and 6B). The reduction of the number of I1 nodes observed in *tfl1* was completely suppressed in *fd-3 tfl1-18* (Fig 6C), while the terminal flower was still formed but much later in inflorescence development than in *tfl1* single mutants. Interestingly, *areb3-1* weakened the *tfl1-18* mutant phenotype in terms of the number of leaves at bolting and the number of I1 nodes, but not in the number of floral nodes formed before the terminal flower phase (Fig 6A and 6C–6E). A further reduction in the severity of the *tfl1* phenotype was observed in *tfl1-18 fd-3 areb3-1*

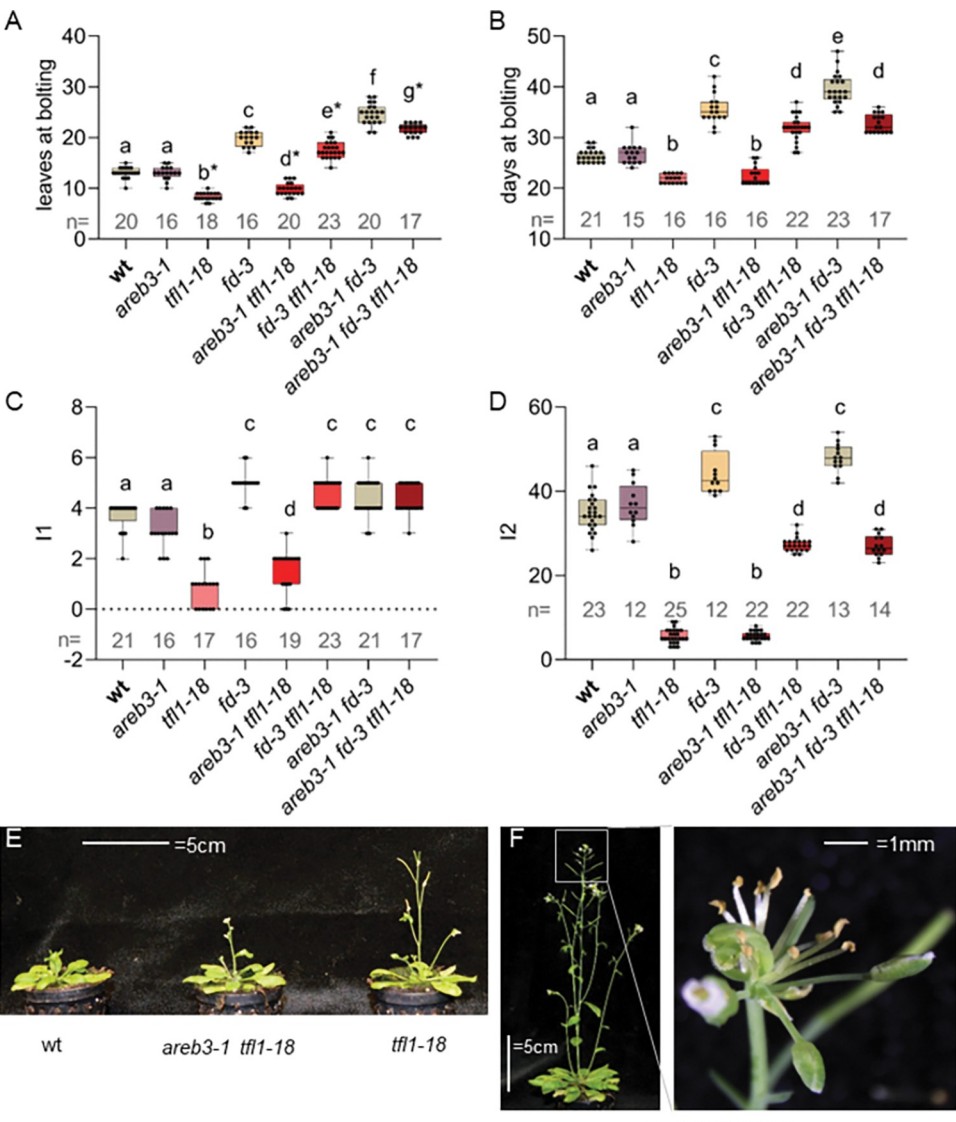

**Fig 6. Genetic interaction between *AREB3*, *FD*, and *TFL1*.** (A) Flowering time of the indicated genotypes under LDs. Asterisks indicate a terminal flower on the main inflorescence. Significance a *vs* b p = 1e-14; a *vs* c p 1e-15; a *vs* d p<5.07e-10; b *vs* d p = 0.031; a *vs* e p = 2e-15; c *vs* e p = 3.15e-05; e *vs* g p = 2,1e-14; f *vs* g p = 2.44e-08. (B) The flowering time of the genotypes indicated in A expressed in the number of days between sowing and bolting. Significance a *vs* b p = 1.37e-7; a *vs* c p = 1.2e-14; b *vs* d p = 1.0e-14; c *vs* e p = 9.91e-6; d *vs* e p<9.0e-14. (C) Mean number of cauline leaves on the main inflorescence of the indicated genotypes under LDs. Significance a *vs* b p<1e-15; a *vs* c p<0.032; b *vs* c p<1e-15; b *vs* d p = 0.041. (D) Mean number of floral nodes on the main inflorescence. Significance a *vs* b p<5e-14; a *vs* c p<1.44e-6: b *vs* d p = 5e-14. (E) The phenotype of the indicated genotypes at 3 weeks after sowing under LDs. *areb3-1 tfl1-18* bolted slightly later than the *tfl1-18* mutant and presented at least one cauline leaf subtending an inflorescence-like structure. (F) The phenotype of the indicated genotypes at 6 weeks after sowing under LDs. *areb3-1 fd-3 tfl1-18* rescued almost all the inflorescence defects of *tfl1-18* but still presented a terminal flower on the main inflorescence (inset).

plants, which were significantly later flowering compared to *tfl1-18 fd-3*, but still earlier flowering than *fd-3 areb3-1* (Fig 6A). Triple mutants of *tfl1-18 fd-3 areb3-1* were nearly identical to *tfl1-18 fd-3* in terms of inflorescence architecture, producing approx. 20 I2 nodes, before forming a terminal flower (Fig 6D and 6F), suggesting that additional factors might mediate FT signalling. Taken together, the genetic interactions between *AREB3*, *FD* and *TFL1* suggest that

AREB3 acts redundantly with FD to mediate FT signalling in terms of flowering-time induction and early termination of the inflorescence meristem.

## Discussion

FD is only partially responsible for relaying florigenic signals at the shoot apex. Here, we demonstrate a wider range of interactions between FT and group A bZIP TFs and show that AREB3 activity partially explains the different phenotypic impact of *ft* and *fd* mutations on floral transition.

### Different group A bZIP TFs mediate FT signalling at the SAM

AREB3 has been assigned to the ABA-related clade of bZIPs [28,29], which is distinct from the clade containing FD and FDP. This may suggest a widespread interaction between ABA responses and flowering-time regulation through bZIP TFs functions. This clade includes ABI5 and ABF2/3/4, which are also flowering-time regulators, mainly operating upstream of *FT* in leaves [21,33,45]. FD and FDP also directly regulate ABA-related target genes and *fd / fdp* single mutants show general deregulation of ABA responses in seedlings, as indicated by their ABA insensitivity during germination [15,21,26]. *AREB3* mutants do not present similar germination defects, but its close homologue *ENHANCED EM LEVEL* (*EEL)* acts as a regulator of embryo-genesis-abundant genes, which are ABA targets [46,47]. Thus, virtually, most group A bZIP TFs appear to be involved in some aspects of ABA signalling regulation, with flowering-time regulation being proposed as a derived state from an ancient role in ABA responses [26]. Therefore, group A bZIPs could independently regulate ABA and flowering pathways in leaves and at the SAM to regulate specific transcriptional responses. Alternatively, the regulation of ABA responses through these bZIPs might constitute a previously uncharacterised level of control of floral transition at the SAM. These aspects warrant further investigations.

Our study demonstrates that AREB3 is functionally redundant to FD in mediating FT signals. The observed distribution of FD and AREB3 at the shoot apex supports that they can regulate common floral targets (Figs 3 and 4). We also note that while *FD* is expressed at higher levels compared with *AREB3* [48], its expression is mainly confined to the inner region of the meristem [7]. In contrast, VENUS:AREB3 accumulated in nuclei of the shoot epidermis, raising the possibility that in this cell layer, AREB3 may be required for more specialised, yet uncharacterised transcriptional and developmental responses. Similarly, in young stage 1 floral buds, FD is absent [20,26], whereas VENUS:AREB3 was detected in these buds and may therefore have a distinct function at that time.

The SAP motif of FD is phosphorylated at T282 [17], which is essential for FT–FD complex formation in yeast and plant cells [19,49,50] and for the promotion of floral transition [15,19]. Biochemical studies identified a family of CALCIUM DEPENDENT PROTEIN KINASEs (CDPKs) responsible for $FD^{T282}$ phosphorylation [17]. AREB3 can be phosphorylated at the corresponding S residue in the SAP motif [36,51], but the precise signals determining the phosphorylation status of the SAP motif of AREB3 and the kinase activity involved are not known yet. Phosphorylation of the ABF/AREB/ABI5 clade typically occurs in response to ABA by SNF1-RELATED PROTEIN KINASE2s (SnRK2s) to activate ABA-dependent responses. Independent phosphoproteomic studies confirmed the ABA-dependent phosphorylation of AREB3 and related proteins *in vivo* [52–56], and *in vitro* by SnRK2s [54–57]. Still, more work is needed to reveal additional phosphorylation events on AREB3, the role of CDPKs in this process and their interaction with ABA-regulated kinases and phosphatases.

FD and AREB3 require an intact SAP motif to mediate FT signalling, as shown by CRISPR-Cas9 targeted mutagenesis (Fig 2). We also show that mutations in the SAP motif of

FD/AREB3 do not abolish their binding to FT in plant cells, unlike in the yeast system (Figs 1, S3 and S4). In BiFC and co-IP experiments, AREB3 proteins carrying alterations at the SAP motif could interact with FT in the nucleus, similar to the wild-type AREB3 protein. In agreement with these findings, *in vitro* experiments showed that Arabidopsis FD [19] and rice FD-like proteins [58] can contact florigen proteins without the bridging function of 14-3-3s. Conversely, rice OsFD1 requires 14-3-3 proteins for *in vitro* interaction with the rice FT-homologue Hd3a protein [14]. Similarly, in tomato, SAP motif mutant alleles of an *FD-like* gene failed to stably retain the FAC in the nucleus, as these FD-like mutated proteins could not interact with the 14-3-3 [49]. The discrepancies between these experiments may be explained by their heterologous nature, including the one on AREB3, in which the bZIP and the florigen proteins were overexpressed in tobacco leaves. Nevertheless, the observation that FT is still bound to AREB3 in nuclei despite the mutation in the SAP motif, suggests that the function of the SAP motif may be biologically separate from the interaction with the florigen protein.

### Functional redundancy and compensatory events at the SAM by bZIP TFs

TFL1 and FT antagonistically regulate floral transition and meristem determinacy of the shoot apex, shaping inflorescence architecture including number, position and identity of lateral primordia [59]. To understand the contribution of AREB3 to this regulatory network, *areb3* mutants were crossed with *tfl1* mutants or *pGas1:FT* plants (Figs 5 and 6), two early-flowering genotypes characterised by over-activity of FT [3,27,42]. Both assays support the redundancy between AREB3 and FD to mediate FT signalling. Similar functional redundancy was previously identified between FD and FDP [26,27]. We also identified genetic redundancy among *FD*, *AREB3* and *FDP* (Figs 5C and S11), and their additive effect in flowering activation supports a model in which FT signals at the SAM can be relayed through multiple FT-bZIP interactions guiding FAC formation to a set of common floral targets. Based on the common evolutionary origin of group A bZIPs [28,60], it is plausible that, as a result of gene duplications, different paralogues have retained similar patterns of expression, and gained or lost new ones. AREB3 broadly retained an *FD-like* pattern of expression at the SAM while this was quantitatively reduced. Increased *AREB3* expression occurs in *fd* mutants, thereby exemplifying a quantitative compensatory mechanism for flowering activation. Compensatory mechanisms derived from positive transcriptional control of paralogues can explain a large proportion of variations in SAM shape and size in Solanaceae [61]. Because of the general conservation of FD-like functions in flowering plants, the contribution of this mechanism to flowering time variability in other species should be investigated. Nonetheless, our findings uncover AREB3 as a novel node of regulation of FT signalling at the shoot, paving the way for a systematic study to elucidate the individual contributions of each bZIP component in decoding FT signals.

## Materials and methods

### Plant material and growth conditions

All genotypes described in this study are in the Columbia-0 (Col-0) background. A list of the genotypes used, and their origin is detailed in the S1 Table. Seeds were stratified for 3–5 days before sowing onto the soil. Plants were grown under controlled conditions in a growth chamber with a mean temperature of 23°C, under LD (16 h light/8 h dark) or SD (8 h light/16 h dark) photocycle. Under LDs, plants were grown under cool white fluorescent tubes (Osram Lumilux Cool White 36 W/840) with fluency of 120 micro-Einstein (Photosynthetically active radiation, PAR). Under SDs, the light was a mix of metal-halide lamps (Sylvania) and fluorescent tubes (300 micro-Einstein PAR). T-DNA insertion alleles of *AREB3* (*areb3-1* –

SALK_061079—and *areb3-2* –SALK_204251) were genotyped according to the SALK SIGnAL instructions [62]. Sanger-based sequencing revealed that both lines contain the same tandem T-DNA insertions, oriented LB-RB-RB-LB (S3 Fig). The *pFD:VENUS:FD fd-3* reporter line was previously described [26].

## CRISPR-Cas9 mutagenesis

CRISPR-Cas9-based mutagenesis on *FD* and *AREB3* genomic sequences was done using the pKI1.1R vector (Addgene Cat. # 85808) following the plasmid depositors' instructions [63]. gRNAs were designed using CHOPCHOP v3 [64] to target the third exon of both *AREB3* and *FD* (S3 Fig), cloned and verified by Sanger sequencing. Oligonucleotides used to assemble the plasmid, are listed in the S2 Table. Engineered vectors *pKI1.1R_AREB3* and *pKI1.1R_FD* were used to transform *fd-3* and wild-type plants, respectively. T1 transgenic seeds were selected by visualizing RFP expression in the seed coat under a Nikon SMZ18 stereomicroscope. Cas9-free T2 mutant plants were counter-selected by isolating non-fluorescent seeds. Mutant alleles were identified by Sanger sequencing and when needed, the chromatograms were deciphered using TIDE [65]. The gRNA designed to produce *areb3-Cr* alleles has no predicted off-targets according to CHOPCHOP, whereas the gRNA used to produce the *fd-Cr* mutants has one predicted off-target (with 3 mismatches) in the coding sequence of *AT2G42230*, an uncharacterised gene. Given the similarity with *AREB3*, we verified by sequencing the absence of mutations in *FDP* and *EEL* coding sequences in homozygous T3 *areb3-Cr* lines and *AT2G42230* in T3 *fd-Cr* lines.

## Generation of transgenic lines

The *pAREB3:VENUS:AREB3* and *pAREB3:VENUS:AREB3*$^{\Delta SAP}$ constructs were generated with the multi Gateway protocol (Invitrogen). Briefly, the *AREB3* promoter (*pAREB3*, 2,797 bp fragment upstream of the *AREB3* start codon) was cloned as 5' element pL4-*pAREB3*-R1 entry vector, the *AREB3* genomic region (1699 bp) from the start codon to the 3'UTR was cloned as 3' element pR2-gAREB3-L3 entry vector and the VENUS sequence as the central element in the pENTR-D TOPO. The pR2-gAREB3-L3 entry vector was mutagenised using oligonucleotides that excluded the SAP motif (RTSSAPF) sequence followed by recircularization of the PCR-produced linearised plasmid using T4 DNA ligase (Thermo Scientific) to obtain the pR2-gAREB3$^{\Delta SAP}$-L3 entry vector. The entry vectors were recombined into pB7m34GW [66], introduced into *Agrobacterium tumefaciens* strain GV3101, and transformed in *areb3-1* (*pAREB3:VENUS:AREB3*) or *areb3-1 fd-3* (*pAREB3:VENUS:AREB3*$^{\Delta SAP}$) backgrounds. 30 T1 BASTA resistant *pAREB3:VENUS:AREB3* independent lines were recovered and 4 T2 lines showing a 3:1 ratio for BASTA resistance were selected for subsequent analysis. Of the 7 independent T1 lines *pAREB3:VENUS:AREB3*$^{\Delta SAP}$, three T2 lines showing a 3:1 ratio of BASTA resistance were analysed for complementation.

The genomic locus of *FD* (3.8 kb upstream of ATG to 1.9 kb downstream of the stop codon) was amplified from Col-0 gDNA and the *3xHA:mCHERRY* tag was inserted by overlap PCR before *FD*'s start codon (S2 Table). The *pFD:3xHA:mCHERRY:FD* fragment was then cloned into the binary vector PER8-GFP after SpeI and XhoI digestion using the In-Fusion HD Cloning Kit (Takara Bio). The final plasmid was introduced into *Agrobacterium* as previously described, and *fd-3* mutant plants were transformed. T1 seeds were sterilized and sown on MS plates supplemented with hygromycin, lines showing a 3:1 segregation were retained, and three homozygous single-copy T3 lines from independent T1 events were selected for further use. After phenotyping, the *pFD:3xHA:mCHERRY:FD fd-3 #7.2* line was used in further assays.

## RNA extraction and qRT-PCR

RNA was purified using the TRIzolReagent (Thermo Scientific) following the producer's protocol. Plant material was ground using the TissueLyser (QIAGEN) bead mill. RNA integrity was checked on agarose gel and it was quantified with the NanoDropOne (Thermo Scientific). After normalization, 500ng of RNA per sample was used immediately for cDNA synthesis (Maxima First Strand cDNA Synthesis Kit for RT-qPCR, Thermo Scientific) and any surplus was stored at -80˚C. Real-time qPCR was performed in a Bio-Rad CFX96Real-Time System with the Maxima SYBR Green qPCR Master Mix (2X, Thermo Scientific) using 15ng of relative RNA template per sample and following the producer's specifications for the reaction. Primers for the gene *IPP2* [67] were used as the internal reference gene, all the gene-specific primers are listed in the S2 Table. RT-PCR was performed using DreamTaq PCR Master Mix (Thermo Scientific) according to the producer's protocol. Experiment-specific plant material growth conditions are described in the main text and the relative figure captions.

## Protein extraction and immunoblot

Total protein extraction was carried out according to published protocol [68] with minor modifications. Briefly, 10–12 SAMs per sample were grounded in liquid nitrogen and homogenized in 50 μL of E buffer (125 mM Tris-HCl pH = 8.8, 50 mM $Na_2S_2O_5$, 1% w/v SDS, 10% v/v Glycerol, 1% v/v HALT Protease Inhibitor Cocktail (Thermo Scientific)). Protein extracts were centrifuged at 15000g for 10 minutes and supernatants were collected. Protein concentrations were quantified using the Pierce Detergent Compatible Bradford Assay (Thermo Scientific) according to the manufacturer's instructions. Samples were diluted with the appropriate amount of 4X NuPAGE LDS Sample Buffer (Thermo Fisher Scientific). Immunoblot analysis was carried out using rabbit αGFP antibody (1:2000; ab6556, Abcam) to detect VENUS: AREB3, rabbit αUGPase antibody (1:2000; AS05086, Agrisera) to detect UGPase, and goat αRabbit-Peroxidase (1:10000; A0545, Sigma-Aldrich, Merck) as secondary conjugated antibody.

## Confocal microscopy analyses

Meristem preparation for confocal microscopy was carried out as previously described [69] with minor modifications [70]. Confocal laser scanning microscopy (SP8; Leica) was performed using settings optimised to visualise VENUS (laser wavelength, OPSL 514 nm; detection wavelength, 521 to 541 nm), mCHERRY (laser wavelength, OPSL 552 nm; detection wavelength, 595 to 621 nm) and Renaissance 2200 (laser wavelength, Diode 405 nm; detection wavelength, 424 to 478 nm).

## Y2H experiments

The complete coding sequences of *FD*, *FDP*, *ABF1*, *ABF2*, *ABF3*, *ABF4*, *EEL*, *GBF4*, *bZIP13*, *FT*, and *TFL1* were amplified with high-fidelity enzymes (New England Biolabs) and cloned into the pDONR201 vector [71] using the Gateway system (Thermo Fisher Scientific). *ABI5* (TOPO_U06_B06), *AREB3* (TOPO_U14_F04), *DPBF2* (TOPO_U03_D06), and *bZIP15* (TOPO_U01_B01) cloned into the pENTR/D-TOPO vector were obtained from the Arabidopsis Biological Resource Center. Gene-specific primers (S2 Table) were used to mutate the codon that encodes the phosphorylatable amino acid of the SAP motif of AREB3 (S294A) or FD (T282A) into alanine, resulting in a non- phosphorylatable SAP motif. The resulting sequences were cloned into pDONR201. The bZIP genes were recombined into pDEST22 (Activation Domain–AD), whereas *FT* and *TFL1* were cloned into pDEST32 (DNA-Binding

Domain–BD; Thermo Fisher Scientific). All plasmids were confirmed by sequencing. Protein-protein interactions were tested using the Y2H system. Plasmids were co-transformed into the yeast PJ69-4A strain following the Frozen-EZ Yeast Transformation II (Zymo Research) protocol. Co-transformation selection was carried out in SD plates lacking the leucine and tryptophan amino acids (–L–W). Three to six colonies were randomly selected, mixed in distilled water, and plated on SD plates lacking–L–W or leucine, tryptophan, and histidine (–L–W–H). Yeasts were grown at 30˚C for six days before image acquisition.

The plasmids used in the Y2H in S2 Fig were obtained by recombining the entry vectors of *FD* and *AREB3* into the pGADT7-GW (AD) destination vector [72], while *AREB*$^{\Delta SAP}$ was recombined from the previously obtained pR2-gAREB3$^{\Delta SAP}$-L3 entry vector into the pDONR201 vector. *FT* entry vector was cloned into pGBKT7-GW (BD) [72] and the Y2H was performed as described previously [58].

## Bimolecular fluorescence complementation (BiFC) assay

Using the same entry vectors prepare for the Y2H experiments, *FT* cds was cloned into pB4GWcY (AB830555.1) and *AREB3*, *AREB3*$^{\Delta SAP}$, or *HDA19* [73]–the latter used as a negative control–were cloned into pB4nYGW (AB830552.1). BiFC backbone plasmids were obtained from the Shoji Mano group and the BiFC protocol was performed as described [74].

The nuclear marker *pUBQ:H2B:mCHERRY* was provided by Eirini Kaiserli (University of Glasgow, UK). *Agrobacterium* cultures carrying the plasmids of interest were grown and a mixture of suspensions of $OD_{600} = 0.2$ each for the BiFC plasmids and $OD_{600} = 0.1$ for the mCHERRY nuclear marker plasmid were co-infiltrated into 2-week-old *N. benthamiana* leaves. The semiquantitative analysis of BiFC interactions [75] was performed by measuring the average fluorescence signal of the nuclei expressing the interacting proteins (min four z-stack images of n>150 nuclei in two independent replicates) using Nikon NIS-Elements software. Absolute YFP and mCHERRY signals were measured and YFP/mCHERRY ratio was calculated.

## Tobacco co-IP assay

The pAM backbone [76] was used to recombine wild-type and mutated versions of *AREB3* and *FD* with N-terminal *5xMyC*, whereas *FT* and *TFL1* were recombined with C-terminal *GFP*. CO was used as a negative interactor of FT. For that, its complete coding sequence was amplified, cloned into pDONR201 and later recombined with N-terminal *5xMyC*. All obtained plasmids were transformed into *Agrobacterium*. Co-infiltration of *N. benthamiana* leaves, protein extraction and IP (GFP-trapA, Chromotek) of GFP-tagged proteins were performed as previously described [77]. Western blotting was performed to detect pulled-down proteins using an αGFP antibody (ab290; Abcam) and co-immunoprecipitated with an αMyC antibody (9E1; Chromotek). Chemiluminescence detection of proteins was performed using the Chemi-Doc MP Imager and associated chemistry (Bio-Rad).

## Analysis of flowering time

To analyse flowering time, plants were grown on soil:vermiculite:perlite mix (3:1:1) in a controlled environment room under LD or SD conditions. Flowering time was measured by scoring the number of rosette leaves, excluding cotyledons, in randomized experiments. When indicated, data regarding cauline leaf number (I1 phase), silique number on the main stem (I2 phase), days to bolting (days between sowing and a visible bolt initiation), and days to flower opening (days between sowing and the opening of the first flower) were also recorded.

## Supporting information

**S1 Table. AREB3 phosphorylation site list, obtained as PhosPhAt4.0 output for AT3G56850.**
(XLSX)

**S2 Table. Oligonucleotides used in this work.**
(XLSX)

**S3 Table. Numerical data that underlies graphs and summary statistics.**
(XLSX)

**S1 Fig. Group A bZIPs interactions with FT and TFL1.** Y2H assays testing protein interactions among group A bZIP TFs and the PEBP proteins FT and TFL1. The panel present in Fig 1 was extracted from the 1:1 dilution column.
(TIF)

**S2 Fig. AREB3$^{\Delta SAP}$ interactions with FT in Y2H assays.** Y2H assays testing protein interactions between FT and the bZIP TFs FD and AREB3. In the AREB3$^{\Delta SAP}$ construct, the SAP motif was completely removed and a stop codon was inserted after R290 (R291*). AREB3 interaction with FT is weakened but not suppressed by the lack of the SAP motif.
(TIF)

**S3 Fig. Complementary tobacco co-IP assay.** *N. benthamiana* co-IP of protein interactions among wt FD and FD$^{\Delta SAP}$ with FT. CO was used as a negative control for the interaction with FT. Pairwise protein–protein interactions were tested by co-agroinfiltration of tobacco leaves. CO and both FD versions were translationally fused to MyC, whereas FT was translationally fused to GFP. The input was composed of total proteins recovered before the IP. GFP-fused proteins were pulled down using anti-GFP nanobody (VHH) beads and immunoblotted using anti-MyC or anti-GFP antibody.
(TIF)

**S4 Fig. BiFC assay showing AREB3 interaction with FT.** (A) BiFC assays testing protein interactions between AREB3 or AREB3$^{\Delta SAP}$ and FT. *N. benthamiana* plants were co-infiltrated with vectors expressing mCHERRY-tagged nuclear protein H2B as a nuclear marker, C-YFP-FT, and either one of N-YFP-AREB3, N-YFP-AREB3ΔSAP or the nuclear protein HDA19 as negative control. Z-stack maximum projections are shown in the pictures. Scale bar 100μm. (B) Semiquantitative analysis of BiFC interactions. The average fluorescence of the nuclei expressing interacting proteins was quantified on n>150 nuclei in two independent replicates. Absolute YFP and mCHERRY signals were measured and the YFP/mCHERRY ratio was calculated. No significant differences were observed between AREB3 and its truncated version AREB3$^{\Delta SAP}$ (R291*) lacking the SAP motif.
(TIF)

**S5 Fig. The flowering phenotype of *areb3* and *fd* mutants.** (A) Days at bolting of the indicated genotypes grown under LDs, measured as number of days between sowing date and the date in which the plants presented a >5mm high floral bolt. Significance a *vs* b, c p = 4.89e-10; b *vs* c p = .89e-10 (left panel); a *vs* b, c p<2.63e-6; b *vs* c p = 1.67e-11 (right panel). (B) Days at bolting of plants grown in SD, data analysis revealed no significant differences. (C) Number of cauline leaves (I1 phase) of plants grown under LDs. Significance a *vs* b p = 4.89e-10 (left panel); d *vs* e, f p<4.17e-7; e *vs* f p = 1.36e-8 (right panel). (D) I1 of plants grown under SDs, revealed no significant differences.
(TIF)

**S6 Fig. *AREB3* gene structure and T-DNA lines used in this study.** (A) Schematic representation of the *AREB3* gene, the position of T-DNA insertions (and corresponding SALK identifier), and the position of the primers used (green). (B) Agarose gel showing amplification of genomic DNA extracted from wt, SALK_061079 and SALK_204251 homozygous plants. Note that both SALK lines show the same amplification pattern using primers on T-DNA left border (LBb1.3) and primers both upstream (mr242) and downstream (dm42) of the putative insertion side. This shows that SALK_061079 and SALK_204251 lines contain the same T-DNA insertions. (C) Sequencing of the amplicons using primers mr242 and dm42, respectively, allowed the fine mapping of the T-DNA insertion site. The T-DNA is inserted between Chr3:21046624 and Chr3:21046636 in the intron 2 of the *AREB3* cds and does not affect any exon. Images produced using Geneious version 2022.0 created by Biomatters.
(TIF)

**S7 Fig. Full-length transcript analysis in T-DNA and CRISPR mutants.** RT-PCR on cDNA from a pool of 5 whole seedlings grown for 2 weeks in 1/2MS agar plates under LD conditions. The primers used (mr244-mr245) amplify the whole cds. Residual accumulation of full length *AREB3* transcript is visible in the double mutants of *areb3-1 fd-3* and *areb3-2\* fd-4* starting from 30 cycles of amplification. At 40 cycles, a residual expression is visible as a faint band also in single *areb3-1* and *areb3-2\** mutants. Similar results were obtained using a primer pair that amplifies a fragment starting at the 3'-terminal portion of the 1st exon and ending 124bp into the 3'UTR (art90-dm42).
(TIF)

**S8 Fig. *AREB3* CRISPR alleles.** (A) Schematic representation of the third *AREB3* exon with the position of the PAM sequence (blue), gRNA target (light blue) and the SAP motif (purple) (B) Genomic sequences of the CRISPR mutants isolated and their predicted protein sequences. The SAP motif sequence is highlighted in purple, the gRNA target sequence is underlined. 5 independent lines were isolated, and two more alleles were detected in T1 plants (see S9B Fig) but not isolated as homozygous lines (n.i.). The altered aminoacidic sequence of the mutants due to the frameshift mutation is highlighted in grey. Images produced using Geneious version 2022.0 created by Biomatters.
(TIF)

**S9 Fig. Isolation of *AREB3* CRISPR alleles.** (A) $T_1$: Flowering time expressed as number of rosette leaves of transgenic, RFP-selected $T_1$ *AREB3* CRISPR lines (*areb3-Cr_T1 fd-3*) compared with transformed non-transgenic RFP- control lines (*areb3-Cr_CTRL- fd-3*). (B) 21 $T_1$ individuals genotyped by sequencing and categorized into the following groups: *areb3-Cr_T1_FSh fd-3* (frameshift mutations); *areb3-Cr_T1_Het fd-3* (heterozygous lines with one wt allele); *areb3-Cr_T1_NE fd-3* (transgenic, not edited lines, both alleles are wt); *areb3-Cr_T1_InF fd-3* ($T_1$ line with an in-frame, -6 deletion in heterozygosity with a wt allele). (C) $T_2$: Flowering time of RFP-, Cas9-free $T_2$ independent *AREB3* CRISPR mutant lines. Random individuals were genotyped by sequencing: *areb3-Cr1_-1/+T fd-3* is a segregating population of biallelic frameshift mutants (-1/+T); *areb3-Cr2 fd-3* is homozygous for a single nucleotide insertion (+A/+A), *areb3-Cr3_-6/+T fd-3* is heterozygous for an in-frame, -6 deletion and a single nucleotide insertion (+T), and *areb3-Cr5_Het fd-3* is heterozygous for a single nucleotide insertion (+G) and a wt allele (significance a *vs* b p = 8e-15; b *vs* c p = 6.67e-5; c *vs* d P<4.38e-7; p value on heterozygous populations not calculated). (D) The phenotype of *areb3-Cr1* and *areb3-Cr2* lines, obtained by backcrossing of the parental *areb3-Cr1 fd-3* and *areb3-Cr2 fd-3* lines with the wt at 4 weeks after sowing under LDs. (E) Flowering time

expressed as number of rosette leaves of *areb3-Cr1* and *areb3-Cr2* lines (significance a *vs* b p<0.0005). (F) Days at bolting and (G) number of cauline leaves (I1 phase) of the indicated genotypes. No significant differences were observed.
(TIF)

**S10 Fig. Expression analysis of *pAREB3*:*VENUS*:*AREB3 areb3-1* transgenic lines.** (A) RT-PCR showing the production of a complete, correctly spliced, transcript in the *pAREB3*: *VENUS*:*AREB3* lines. (B) Real-time qPCR analysis of the same lines. Line 26.9 was chosen for subsequent experiments as the most similar to the expression levels of the wt. In both experiments, plants were grown for 2 weeks in SD (D0) before being moved to LD and resampled after one full day of LD (D1) at ZT8. (C) Flowering time analysis of *areb3-1* mutant complemented with vector expressing either the complete *AREB3* genomic sequence, or its truncated version lacking the last 24nt coding for the SAP motif. pA3VA3 $T_3$26_9 in *areb3-1* is an isogenic *pAREB3*:*VENUS*:*AREB3* line that was introgressed in the mutant *areb3-1 fd-3*. pA3VA3$^{\Delta SAP}$ are independent *pAREB3*:*VENUS*:*AREB3*$^{\Delta SAP}$ $T_2$ lines in *areb3-1 fd-3* (significance a *vs* b, c p<3.82e-4; b *vs* c p<9.09e-3).
(TIF)

**S11 Fig. Expression analysis *FT* transcript by RT-qPCR.** Plants were grown under LD conditions, and the above-ground plant material (rosette and hypocotyl) was collected at ZT16 at 7 days (D7) and 15 days (D15) after sowing, representing pre- and post-floral transition stages. Each point represents a pool of >3 individual plants. Mixed-effect analysis, Tukey's multiple comparisons test evidenced no significant difference in all the tested data, excluding *areb3-1 vs areb3-Cr1 fd-3* (p = 0.0313). One of two biological replicates is shown here.
(TIF)

**S12 Fig. Flowering time analysis of multiple mutants for the group A bZIPs *FD*, *AREB3* and *FDP* genes.** Plants were grown under LD conditions. Significance a *vs* b, c, d, e p<9.97e-06; b *vs* c p = 1.21e-02; c *vs* d p<1e-15; c *vs* e p = 4.52e-10; d *vs* e p = 2.14e-3.
(TIF)

## Acknowledgments

We thank present and former members of the Conti laboratory, in particular Aldo Sutti, Sara Colanero, Alice Robustelli Test, and Giada Lavigna for their help, personnel at Orto Botanico Città Studi for plant care, and Giorgio Perrella for the HDA19 plasmid and help with the BiFC assays. BiFC was carried out at NOLIMITS, an advanced imaging facility established by the Università degli Studi di Milano. We thank Ton Timmers from the Central Microscopy service (MPIPZ) and the members of the Coupland group, especially Martina Cerise, Sara Cioffi and Cristian González Jiménez, for their helpful assistance and fruitful discussions. We thank Maida Romera-Branchat and Chloé Pocard for extensive discussions and some earlier comparisons of the roles of FD and AREB3. Finally, we thank Prof. Rüdiger Simon for his insightful comments and suggestions.

## Author Contributions

**Conceptualization:** Damiano Martignago, Vítor da Silveira Falavigna, Lucio Conti.

**Data curation:** Alessandra Lombardi, Massimo Galbiati.

**Formal analysis:** Damiano Martignago, Vítor da Silveira Falavigna.

**Funding acquisition:** Chiara Tonelli, George Coupland, Lucio Conti.

**Investigation:** Damiano Martignago, Vítor da Silveira Falavigna, He Gao, Paolo Korwin Kurkowski, Massimo Galbiati, Lucio Conti.

**Resources:** He Gao, George Coupland.

**Supervision:** George Coupland, Lucio Conti.

**Visualization:** Damiano Martignago, Vítor da Silveira Falavigna.

**Writing – original draft:** Damiano Martignago, Vítor da Silveira Falavigna, Lucio Conti.

**Writing – review & editing:** Damiano Martignago, Vítor da Silveira Falavigna, Massimo Galbiati, Chiara Tonelli, George Coupland, Lucio Conti.

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
