## [Decision Letter · Decision Letter 0]

19 Jan 2023

Dear Dr Conti,

Thank you very much for submitting your Research Article entitled 'The bZIP transcription factor AREB3 mediates FT signalling and floral transition at the Arabidopsis shoot apical meristem' to PLOS Genetics.

The manuscript was fully evaluated at the editorial level and by two independent peer reviewers. The reviewers appreciated the attention to an important problem, but raised some substantial concerns about the current manuscript. The reviewer 1, in particular, asked for some more data of SAP domain function, to which we agree. Based on these reviews, we will unfortunately not be able to accept this version of the manuscript, but we would be willing to review a much-revised version. 

If you decide to revise the manuscript for further consideration at PLOS Genetics, please aim to resubmit within the next 60 days, unless it will take extra time to address the concerns of the reviewers, in which case we would appreciate an expected resubmission date by email to plosgenetics@plos.org.

We are sorry that we cannot be more positive about your manuscript at this stage. Please do not hesitate to contact us if you have any concerns or questions.

Yours sincerely,

Li-Jia Qu

Section Editor

PLOS Genetics

Reviewer's Responses to Questions

**Comments to the Authors:**

Reviewer #1: In this manuscript Martignago et al., demonstrated the role of a transcription factor AREB3 in flowering time control by mediating FT signaling. The research started through a Y2H assay screening group A bZIPs that interact with florigen protein FT and TFL1. They found almost all of them interact with FT and TFL1 and have a SAP domain that has been shown functionally important in FD, a well know group A bZIPs TF in FT signaling. However, they continued with AREB3, the only one showing a canonical SAP except for the known FD and FDP. By genetic study and confocal imaging analysis they showed the functional redundancy of AREB3 with FD and FDP in FT signaling and TFL1’s antagonizing FT. The authors also generated a series CRISPR mutants of AREB3 to disrupt the SAP domain, which give strong genetic evidence supporting the functional importance of the SAP domain, however the biochemistry experiments didn’t show any influence from SAP deleted AREB3 on its physical interactions with FT and TFL1.

This is an interesting story which provides important clues regarding how hormones such as ABA is involved in flowering time control. The genetic analyses for the redundancy of between ABRE3 and FD& FDP and ABRE3’s function in FT signaling are fairly complete and great. However, whether the SAP domain is involved in this FT signaling is not clear. Therefore, I would expect more analyses on SAP domain function which will strengthen the conclusions made by the authors. The detailed comments are listed below.

1. The narrative order about the SAP domain is a bit confusing. I suggest the assays about the SAP domain including both the genetic and biochemical parts (line 193-211 & line 149-172) to be integrated as an independent session and with some more analysis suggested below to be put in the last session.

2. To make it clear whether the SAP domain is involved in this FT signaling, at least one of the following assays need to be done:

1) The flowering phenotype of either pGAS1:FT areb3-Cr1 fd-3 or tfl1 areb3-Cr1 fd-3 (Cr1 can also be Cr2 or Cr4).

2) The transient assays in tobacco leaves showing deletion of SAP domain doesn’t influence AREB3 FT/TFL1 interactions are not convincing, as the proteins are all overexpressed which may cause false positive. In vivo, the SAP domain may influence the protein stability or localization which might make the spatially co-existing of AREB3 and FT/TFL1 totally impossible. To this end, a VENUS:AREB3-ΔSAP is necessary to show both the abundance and spatial distribution of AREB3-ΔSAP is still the same with the WT AREB3. If not, the transient assay wouldn’t be useful to explain how SAP domain function.

3. It would be better to add some discussion about the canonical and noncanonical SAP domain. Any functional differentiations between?

4. Line 256-261, is there any evidence that ABRE3 protein level always consists with mRNA level during development? What makes it more unconvincing is that the ABRE3 mRNA is not detected in abre3-1 (no phenotype) but in abre3-1 fd-3 (additively late flowering). So the mRNA level may not be sufficient to explain the compensation for FD loss. Either a western blot for protein level of ABRE3 in fd-3 or quantification of fluorescence in figure3 is necessary.

5. Some data are missing or wrongly cited:

1) Line 213, no corresponding data were found in Figure S9, it seems should be S7, but also lack in S7.

2) Line 237, wrong citation, should be S9.

3) Line 293: citation is missing.

6. Some figures and legends need to be improved to make is clearer:

1) Figure 1D, GFP:IP panel seems overexposured, could the it be performed with shorter exposure so the band is visible? Besides, is there any explanation why the sizes of the IPed bands of Myc-AREB3 and Myc-AREB3 S294A are different?

2) Figure 1B, could the phosphorylated sites T282 and S294 indicated in the amino acid sequence?

3) Figure S6 looks a bit mess and the complicated names of primers makes it so difficult to follow with the details. Strongly suggest to make it clearer.

4) Figures 3 & 4, what are 11 LD, 13 LD and 18LD FM et al. mean?

Reviewer #2: The manuscript by Martignago, da Silveira Falavigna, et al., entitled “The bZIP transcription factor AREB3 mediates FT signalling and floral transition at the Arabidopsis shoot apical meristem” reports the characterization of a FD paralog in Arabidopsis, AREB3, that can interact with FT and TFL1. Genetic analysis indicated that areb3 mutation enhanced the flowering phenotype of fd mutants. Floral transition is a topic of enormous significance, and this study reports a new player. They also showed an transcriptional compensation mechanism that has been recently found in plants (Rodriguez-Leal et al., Nat Genet 2019, 51:786–792). Overall, this is a solid study with conclusions supported by the data. I have only some minor comments.

1. P. 7, line 150, “We found that all group A bZIP genes presented at least one splicing form encoding highly conserved SAP domains within a previously described conserved carboxy-terminal region (Fig 1B [28]).” Line 153, “Besides FD and FDP, only AREB3 (also known as DPBF3, AtbZIP66, At3g56850) presented a canonical SAP motif.” It is confusing how AREB3 was selected. It seems that all bZIPs shown in Figure 1B have the SAP domain.

2. The authors hoped to use CRISPR to generate SAP domain deletion lines. However, frameshift may also interfere with gene expression. Are the truncated transcripts still expressed at levels comparable to the full-length version?

3. The roles of the SAP domain is confusing. On one hand, genetic analysis indicated that the SAP domain is required for AREB3 function. On the other hand, molecular assay indicated that SAP is not affecting AREB3 interaction with FT/TFL1. How to reconcile this discrepancy?

**Have all data underlying the figures and results presented in the manuscript been provided?**

Reviewer #1: **No: **As discribed in the comments to the authors

Reviewer #2: Yes

PLOS authors have the option to publish the peer review history of their article (what does this mean?). If published, this will include your full peer review and any attached files.

Reviewer #1: No

Reviewer #2: **Yes: **Yuling Jiao

---

## [Decision Letter · Decision Letter 1]

27 Apr 2023

Dear Dr Conti,

We are pleased to inform you that your manuscript entitled "The bZIP transcription factor AREB3 mediates FT signalling and floral transition at the Arabidopsis shoot apical meristem" has been editorially accepted for publication in PLOS Genetics. Congratulations!

Yours sincerely,

Li-Jia Qu

Section Editor

PLOS Genetics

Comments from the reviewers (if applicable):

Reviewer's Responses to Questions

**Comments to the Authors:**

Reviewer #1: In the revised manuscript the authors have addressed all of my concerns, I now recommend it for publication.

Reviewer #2: The authors have sufficiently addressed my concerns. I have no further comments.

**Have all data underlying the figures and results presented in the manuscript been provided?**

Reviewer #1: Yes

Reviewer #2: Yes

PLOS authors have the option to publish the peer review history of their article (what does this mean?). If published, this will include your full peer review and any attached files.

Reviewer #1: No

Reviewer #2: No

**Data Deposition**

http://datadryad.org/submit?journalID=pgenetics&manu=PGENETICS-D-22-01451R1

**Press Queries**

---

## [Editor Report · Acceptance letter]

11 May 2023

PGENETICS-D-22-01451R1 

The bZIP transcription factor AREB3 mediates FT signalling and floral transition at the Arabidopsis shoot apical meristem 

Dear Dr Conti, 

We are pleased to inform you that your manuscript entitled "The bZIP transcription factor AREB3 mediates FT signalling and floral transition at the Arabidopsis shoot apical meristem" has been formally accepted for publication in PLOS Genetics! Your manuscript is now with our production department and you will be notified of the publication date in due course.

With kind regards,

Anita Estes

PLOS Genetics

On behalf of:
